# Caregivers of Individuals with Cancer in the COVID-19 Pandemic: A Phenomenological Study

**DOI:** 10.3390/ijerph19010185

**Published:** 2021-12-24

**Authors:** Leila Abou Salha, Julio Cesar Souza Silva, Cleusa Alves Martins, Cristiane Soares da Costa Araújo, Edinamar Aparecida Santos da Silva, Angela Gilda Alves, Cácia Régia de Paula, Flavio Henrique Alves de Lima, Veidma Siqueira de Moura, José Elmo de Menezes, Virginia Visconde Brasil, Maria Alves Barbosa

**Affiliations:** 1Faculty of Medicine, Federal University of Goiás, Goiania 74605-050, Brazil; juliocesarsouzasilva@hotmail.com (J.C.S.S.); cleusa.alves@gmail.com (C.A.M.); edinamar@ufg.br (E.A.S.d.S.); lucano945.fl@gmail.com (F.H.A.d.L.); veidmaenf@gmail.com (V.S.d.M.); maria.malves@gmail.com (M.A.B.); 2Faculty of Nursing, Federal University of Goiás, Goiania 74605-080, Brazil; crisarau@yahoo.com.br (C.S.d.C.A.); angelagildaalves@gmail.com (A.G.A.); viscondebrasil@gmail.com (V.V.B.); 3Faculty of Nursing, Federal University of Jataí, Jatai 75801-615, Brazil; cregia@ufj.edu.br; 4Federal Institute of Education, Science, and Technology of Goiás, Pontifical Catholic University of Goiás, Goiania 74605-900, Brazil; jelmo.maf@gmail.com

**Keywords:** COVID-19, cancer, caregivers, qualitative research

## Abstract

Caregivers of individuals with cancer in the COVID-19 pandemic are faced with the demands of cancer and the health needs produced by it, along with their own health and self-care needs, and the uncertainties of expectations and risks. A qualitative analytical phenomenological study with caregivers of individuals with cancer registered at the state referral hospital supplying medications, who answered the sociodemographic assessment questionnaires and semi-structured questions about their feelings and perceptions in the COVID-19 pandemic. Bardin’s content analysis was used, with methodological quality assessed using SRQR Standards for Reporting Qualitative Research and the MAXQDA software. Most of the caregivers are women, married, Catholic, of low income and education, aged between 30 and 60 years, optimistic, comply with health guidelines regarding social distancing, use of masks, and routine hand hygiene, do not practice regular physical activities, mention concern for their own physical and financial survival, and that of their family. The main need identified in the affective sphere was to reframe contact with family members, seeking to strengthen the bonds of affection. The feeling of emotional vulnerability shows the importance of building effective public policies for social support consistent with the improvement of health care for this population.

## 1. Introduction

The COVID-19 pandemic has presented some unique behavior in the world, with the emergence of new viral strains and a significant increase in the number of cases in many countries as well as a reduction in cases in others with more flexibility of care [1]. In Brazil, the new epicenter in South America, the number of new cases and deaths has reached historic records [2,3]. These data represent a profound lack of political and social control that generates uncertainties in expectations and risks, especially in patients with chronic diseases such as cancer [4,5,6].

Individuals with cancer are at high risk in the COVID-19 pandemic. They are vulnerable to infection due to possible immunosuppression, in addition to the increased risk of presenting serious complications if they are infected. As a result, they are neglected in priority care among those infected with COVID-19 [6,7].

Cancer occupies an important place as a global public health problem, with one in six deaths being due to cancer [8]. In Brazil, in the years 2020–2022, it is estimated that there will be approximately 625,000 new cases of cancer (450,000, not considering cases of non-melanoma skin cancer). Of this number, about a third of new cases could be avoided by reducing or even eliminating environmental risk factors and those related to lifestyle habits, such as tobacco cessation, adopting a healthy diet, and practicing physical activities [9].

It is important to recognize the impacts on family dynamics and the need for caregivers in the treatment of people with cancer, which can add quality to the living years, helping maintain functional capacity and autonomy, and giving new meaning to existence [10,11].

The caregiver’s daily life is directly influenced by the demand for care produced by cancer and the patient’s own health needs [12,13,14,15,16,17]. In addition, the stigma of cancer, which is still present, can deprive individuals with cancer and their caregivers of social outlets [18,19,20,21]. These demands have increased due to the public health impositions of social isolation, both for the person with cancer and for their caregiver.

Through a Heideggerian approach, ontological perspective on giving care, this study looked for multiple meanings in the responses of the caregivers, and highlighted senses of being-in-the-world in such proximity to cancer, and the construction of bonds with the individual with cancer [22,23].

Taking care of others is also taking care of oneself, and to accomplish this, it is important to know the other person, their motivations, and needs, in the context of responsibility for oneself as a being-in-the-world [24].

In this sense, the family caregiver of a person with cancer is the object of this study, for which the guiding question was: who are you in the COVID-19 global pandemic? The objective was to analyze the perception of self-care, concerns, and attitudes towards the global COVID-19 pandemic.

## 2. Materials and Methods

### 2.1. Ethical Considerations

This study involved human beings and complied with the ethical and legal precepts regulated by the National Health Council as per Resolutions 466/2012 and 510/2016 [25,26]. It was submitted to the Research Ethics Committee of the Federal University of Goiás and approved on 20 August 2018, under opinion no. 2.831.905 and subsequent amendment approved on 25 February 2021, opinion no. 4.558.046–CAAE 93238318.7.0000.5083.

### 2.2. Study Design

In this study that is a part of a large study as a doctoral thesis in health sciences, the analytical phenomenological study design was used.

### 2.3. Study Setting

The present study was conducted with caregivers of cancer who were registered users of a centralized state drug supply service in Goiânia, Goias, Brazil, from March 2020 to March 2021.

The phenomenological study aims to understand individual demands through the analysis of experiences and the meanings of thereof, that reflect on care, the role of the caregiver, and the patient. Utilizing the existential phenomenology studied by Martin Heidegger allows for a deeper understanding of being, its languages, and experiences in the phenomenon studied [27].

The investigation considered the guiding question: who are you in the COVID-19 pandemic?

### 2.4. Participant Selection and Data Collection Procedure

The selection of participants included registered users of the state referral service for dispensing cancer drugs in 2020. They were invited in person and by telephone, in March, July, and November 2020 (3 successive invitations), obtaining the sample by saturation in March 2021. Participants received, through a messaging application and/or e-mail, the survey instrument created with the Google Forms tool [28], as well as the Free and Informed Consent Form, which contains information about the purpose of the research, the type of participation desired, and the probable time of completion and agreement to this was a prerequisite for accessing the research instrument. Participants were identified by codes (C1 to C42) for anonymity purposes.

The inclusion criteria included being a caregiver of a person with cancer and using oncological medications. The study excluded drug users, relatives who were not caregivers, and people registered at the dispensing service who were not caregivers, such as users’ lawyers and drivers.

The main researcher is an observer of the medication dispensing process at the service in question. The process was subjected to an external audit to monitor and validate the selection process.

The data collection instrument was created by the researchers in the form of a questionnaire and included a sociodemographic assessment with eight questions on the familial relationship with the user of the oncological drug, age, gender, education, religion, marital status, number of people living in the home, and family income, in addition to five semi-structured questions that addressed self-care (the practice of physical activity, type of activity, frequency of exercise, and place where exercise is practiced), concerns (financial and emotional), perceptions (physical and emotional health), and attitudes (compliance with health rules and what care practices they perform) of informal caregivers in the face of the COVID-19 pandemic.

### 2.5. Data Analysis

Sociodemographic data were analyzed to obtain means, standard deviations, and respective 95% confidence intervals.

The responses to the semi-structured questions were analyzed using thematic content analysis [29]. The technique is divided into chronological steps: pre-analysis; exploration of the material; treatment of results; and interpretation of data. After data analysis, the responses were categorized using MAXQDA 2020 software [30] and presented following the Standards for Reporting Qualitative Research (SRQR) checklist [31] from the information presented by the caregivers.

MAXQDA, version 2020, is a software program for analyzing qualitative data used in content analysis and allows the creation of codes and categories and the association of variables [30].

The SRQR is a checklist with 21 items, used in qualitative research, that improves methodological quality by providing transparency and comprehensiveness to the articles [31].

The original contributions presented in the study are included in the article, further inquiries can be directed to the corresponding author.

## 3. Results

Of the total of 365 registered patients, the inclusion and exclusion criteria were applied, 42 caregivers were selected who responded to one of the three invitations that made up the study sample. The speeches were categorized and used the MAXQDA tool, 2020 version [30], which grouped common points and quantified the results according to frequency, four ontological themes emerged from this analysis.

The study included 42 caregivers, 38 (90.5%) who considered themselves relatives and 04 identified as conservator/legal guardians (9.5%), women (55%), men (45%); age group between 30 and 60 years; marital status with a predominance of being married (76%); predominantly Catholic religion; elementary school education (62%); with more than three people in their household (58%), and a family income up to the minimum wage (62%) (Table 1).

The responses revealed the perceptions of caregivers of people with cancer during the COVID-19 pandemic, and gave rise to four ontological themes:

### 3.1. Sacrificing Self-Care Due to Work Demands

Most of the caregivers of cancer patients recognize that they do not practice physical activities (67%), due to being overloaded with housework or physical labor. Those who can practice physical activities (33%), perform them outdoors 3 to 4 times a week.


*It’s impossible to do any activity because you can’t leave the house*

*(C1)*



*I do not practice specific physical activities, only those related to housework and those related to remote work.*

*(C17)*



*I work most of the time, I don’t practice much activity because I take care of my father.*

*(C35)*



*Not very active. Walking. Around the block. Almost every day (3 to 4 times a week).*

*(C42)*


### 3.2. Worrying about the Future: Suffering and Unemployment

The responses reveal a greater concern with health issues (76%) than financial issues (12%), with some concerned with both issues (5%).

Regarding health issues, they discussed fear of their own suffering and that of their families (77%), and the fear of death (5%). On the topic of finances, the fear was of unemployment and not providing financial support for their family (18%).

Talking about their own health in the future, most are optimistic because they believe the disease will eventually regress (26%), because they trust science to develop vaccines and effective treatments (21%) and because of religious influences such as faith in better days (17%).


*My concern is about getting sick and not being able to take care of my father.*

*(C4)*



*…how I can survive without income…*

*(C13)*



*…staying healthy and taking care of emotional and financial health.*

*(C17)*



*…with my mother’s health! Fear of her contracting the virus because her immunity is low due to her treatment!*

*(C33)*



*…always hoping to improve. Because we trust in God always to give us strength.*

*(C5)*



*…get worse. We get worse every day in this life.*

*(C12)*



*I hope to improve with the end of my son’s chemotherapy.*

*(C26)*



*Look, I have faith in God that my health will improve in Jesus’ name.*

*(C28)*



*…and the emotional one too, with faith in God, that they will soon find the vaccine for this virus.*

*(C30)*


### 3.3. Complying with Public Health Rules as Closely as Possible

When discussing the official public health guidelines, most recognized that they can follow them (67%), concerning social distancing, isolation, routine hand hygiene with sanitizing gel, and use of protective face masks. Caregivers who admit to partially following the recommendations (28%) cite the difficulty in using daily public transport and the need to go to work as their main reasons.


*I need to come and go all the time, but I take care of myself.*

*(C19)*



*In part, we are very careful to use a mask, hand sanitizer, hygiene, but I go to hospitals a lot.*

*(C26)*



*I started following all the protocols, but I had to keep going to work, even knowing that the number of infected and dead in my own service was getting closer to me.*

*(C31)*



*I’m avoiding crowding, but social isolation is a little difficult to meet due to my service.*

*(C33)*


### 3.4. Positive Actions and Reclaiming Feelings in the Post-Pandemic World

For this topic, the responses addressed the first action that the participants would like to take when the pandemic is brought under control. A hug and contact with people close to them were the desire of most caregivers (43%), followed by trips and vacations (24%), the return of routine life (16%), the need to be thankful/thank God (10%) and the search for a new job (7%).


*Give a hug to my children who live far away from me, I miss them a lot.*

*(C1)*



*Carefully go back, due to low immunity, to having contact with my family.*

*(C2)*



*Embrace the people I love! Including my mother who, due to the disease, we are taking great care to avoid, I will hug her a lot.*

*(C33)*



*Thank God for having passed this challenge.*

*(C40)*



*Thank God for taking care of me and my family, for not having COVID, thank God.*

*(C41)*


## 4. Discussion

COVID-19 as a new disease raises questions about its impact on specific populations, such as individuals with cancer [35]. Cancer is considered one of the main public health problems in the world, considering the set of more than one hundred diseases that have disordered cell growth as a common feature [36]. It is considered that practically one-third of all cancers could be prevented with current knowledge and technologies [8].

Individuals with cancer may be more susceptible to infection by SARS-CoV2, due to the immunosuppression resulting from the treatment and the neoplasm itself, possibly leading to faster worsening of the condition and mortality [35,37]. In the case of contamination, immunosuppressed individuals can remain contagious and spread the coronavirus for more than two months [38,39,40].

In terms of a specific treatment, there is still no consensus in the literature on adequate protocols, maintaining or postponing antineoplastic chemotherapy, and rescheduling surgeries [41,42,43,44,45].

However, continuous treatment of the disease during this period is an essential guarantee in health services, although these services are overloaded with the growing number of cases of COVID-19, requiring medical supplies and qualified personnel. This health scenario can cause inconvenience and delays in treatment and hospitalizations that sometimes negatively affect the prognosis and treatment of the disease [46,47,48].

As a result of this scenario, individual clinical judgment determines the continuity or suspension of cancer therapy in individuals with suspected or confirmed COVID-19 [46].

It is important to recognize cancer as a family disease, both in genetic terms and in the reflexive insecurity caused by the clinical condition, and the care of patients with cancer is characterized by uncertainties [49,50].

Care has been recognized as a formal profession in the national register of occupations since 2011 [51]. However, this study highlights the role of the informal family caregiver, not recognized as a professional and, consequently, with no rights inherent to the formal occupation.

The caregiver is responsible for caring as if it were inherent to their essence [52]. Certainly, care with a scientific and technical basis is not enough to serve people as they demand a sociological approach to care, making interpersonal needs more visible [22].

The guiding question was an invitation to participants to reflect on their experiences [53]. In this sense, perception is considered wisdom since it dialogues with the experiences of the subjects [54].

The concept of being that emerges from the responses indicates daily work overload, anguish, and fear when faced with the uncertainties of the future of the pandemic. The concept suggests ontological movement, being both a caregiver and a relative of the patient, with both unique and routine aspects [23].

Care is one of the foundations of human existence, understood as a basic existential ontological phenomenon, it is a way of being-in-the-world that is structured in human relationships with all things [55].

Taking care of a being is one possible choice when determining how and what to care for, as a way of existing in the world and being part of it as both a singular and plural concept. This sense is expressed in states of mind in which one takes care of existing, identifying the needs and motivations that lead to wanting to care [27,56].

The profile of adult women as informal caregivers is similar to other studies in this area [57,58,59,60].

The caregiver is usually a member of the patient’s family, usually wives or daughters, who embrace care out of necessity, but then stack it with work activities, negatively impacting their quality of life. Most caregivers generally quit their job and become full-time caregivers, greatly compromising the income of the family group, already affected by care expenses [59,60,61,62].

One of the consequences of the pandemic is the increase in unemployment. The official unemployment rate in the last quarter of 2020 was 13.9% [63].

Unemployment results in an increase in jobs moving to the informal market, outsourced workers, subcontractors, flexible workers, and part-time workers. This is a significant group in society that demands a social protection network with public policies to fight hunger and poverty [63,64,65,66,67].

Economic stability has also been negatively impacted, with productive activities stopped, workers lost their formal jobs and livelihoods, resulting in an increase in informality in the economy to the current level of 40%. This has resulted in an increase in loan defaults as well as the cancellation of private health plans and insurance [64,65,66,67,68].

With growing unemployment in the pandemic, many men remained in their homes helping care for their family members, but in the trend towards the division of tasks, the gender logic of care being a naturally female activity remains [59,60].

The pandemic makes the equal division of tasks by family members unfeasible because it limits interaction due to the need for social distancing, exposing conflicts and rifts. On one side, there is forced coexistence and the intensification of care that can lead to an overloaded caregiver, and on the other side, there is emotional isolation and financial difficulties in coping with their own needs along with the needs related to being a caregiver [60,69,70].

The global strategy of using social isolation to prevent the dissemination of the coronavirus, to a certain extent, carries with it a form of violence against workers due to the risk of death when the need for their work is critical [71]. One might think that the pandemic would highlight the humanitarian dilemma of choosing survival and quality of life or saving the economy and production. This is difficult to choose because they are inseparable. Taking care of the population is, above all, the government’s responsibility [72,73].

Individual and collective behaviors have been significantly redefined, and the importance of investing in the production of scientific knowledge has been acknowledged, and these are influences that should not be limited to the duration of the pandemic but become part of our new normal [74].

Taking care of others while taking care of oneself is a life situation faced by the caregiver during the pandemic, given the global health guidelines of social isolation. The national and global media reinforce proper hand hygiene with soap and water or hand sanitizer, along with routine use of protective face masks and social distancing, but despite this, misinformation and false knowledge of COVID-19 are present in a significant portion of the population [75,76]. Fear, to some extent, can be a positive factor for adherence to measures aimed at reducing the spread of the pandemic [77].

The need to go to work and commute by public transportation makes it difficult to follow protective measures in the pandemic. However, it is recognized that social isolation generates anguish and uncertainty that can produce suffering manifesting in depression, loneliness, hopelessness, and fatigue [78].

The being-towards-death stands out for recognizing the proximity to finality and the anguish in mitigating this possibility [79].

It may be difficult for an individual to change their routine and follow protective measures not only from the necessity to work for subsistence but also from the denial of the exceptional reality we are all facing and the attempt to recreate normality—the old normal, to regain their emotional stability [73,80].

In this sense, optimism emerged as prevalent in the responses, demonstrating hope as a positive feeling to minimize negative feelings in coping with this exceptional situation, through trust in science and the emergence of effective treatments, disease mitigation, and spirituality [81].

Caregivers also expressed their desires, which also represent their current emotional and financial needs, in the post-pandemic scenario with implications on the forms of social relationships. In addition to the desire to return to the old normal, a new job, and religious gratitude, the need to strengthen family ties of affection was revealed, with visits to homes, family trips, and hugs, valuing conviviality [81].

The pandemic broke the routine normal patterns of sociability, raising feelings of concern, sadness, and anguish in the face of everything that had been lost [81,82]

Dedication to a sick family member and the distance from other loved ones exacerbated feelings of loneliness and impotence in the face of the coming reality. This fear, arising from the loss of familiarity with the world, generates anguish when confronting this reality due to the strangeness of being in the world [83,84].

The pandemic brought each death closer, redefining humanity under the anthropological prism of fragility and vulnerability [85]. The being-toward-death must lead the individual towards life [27]. However, excessive care and fear generate feelings of perplexity and impotence in the caregiver, who seeks in spirituality a way to strengthen and resist in difficult times [11,50,86]. Thanking God, as an action in the post-pandemic world, is mentioned in many responses.

## 5. Conclusions

The post-pandemic world imposes a new, more unified path, which involves self-care and caring for the other. It also imposes a new perspective, recognizing the need to direct public health, education, and social welfare policies toward this population [78,87].

The present work has strengths and limitations. The size may suggest the generalization of the obtained results. As a possible limitation, the stages of cancer and treatment protocols were not considered relevant, as they do not interfere with the quality of care and were not identified.

## Figures and Tables

**Table 1 ijerph-19-00185-t001:** Sociodemographic profile of the population.

	Absolute Frequency (n = 42)	Relative Frequency (%)
**Gender**		
Female	23	55
Male	19	45
**Civil Status**		
Single	5	12
Married	32	76
Divorced	2	5
Widowed	3	7
**Age**		
20–30	8	19
31 to 60	20	47
61 to 80	14	34
**Years of education ***		
1–9	26	62
10–12	06	14
>12	10	24
**Religion**		
Agnostic	1	2
Catholic	26	62
Evangelical	15	36
**Others living at home**		
None	2	4
One	6	14
Two	10	24
Three or more	24	58
**Family income ****		
Up to 1 MW (up to U$209/mo)	26	62
Between 1–3 MW (Between USD 209–627/mo)	15	36
Over 3 MW (More than USD627/mo)	1	2

* Education: separated into bands according to education level—complete primary level (9 years of study), high school (12 years of study), and higher education (over 12 years of study) [32]. ** Family income: The national minimum wage in force in 2020 was R$1045.00 [33]. Currency quotation 1U$ is equivalent to R$5.00 [34].

## Data Availability

The datasets generated and/or analysed during the current study are not publicly available due to data privacy but are available from the corresponding author upon reasonable request.

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
