# Peer review of "Caregivers of Individuals with Cancer in the COVID-19 Pandemic: A Phenomenological Study"

_ijerph, 2021, doi:10.3390/ijerph19010185_

Round 1

Reviewer 1 Report

I thank the authors for the opportunity to review this interesting article, it is an interesting article that the experience of caregivers of cancer patients. However, it would be interesting to take into account the following recommendations and respond to the questions posed:
• What contact was there between the study participants and the researchers, was it limited by the pandemic situation? How was it done? It would be interesting to make a section of context and study setting
• When collecting data, they talk about questionnaires and semi-structured questions. What questions did they ask? What criteria did they use to make them? How did you request clarification if there was no face-to-face contact with the participants? As stated, there was no two-way communication as could occur in an interview.
• Describe in more detail guaranteed the quality of the study. How they did it?
• I find it interesting that they have the approval of the ethics committee to carry out the study (August 2018) before declaring the pandemic, please clarify this information.
• How did you calculate the sample size?
• In view of the results, the topic of the pandemic hardly appears, only in the last topic, in the first three no mention is made. In addition, the last topic is briefly addressed and therefore does not have so much prominence as to highlight it within the title. Find a title that best fits the content of the article. Similarly, the results obtained deviate from the general objective of the study, it would be interesting if there was a proportional correspondence between the results and the objectives set.
• In the discussion, they affirm that the results could be generalized, even starting from a sample size of 42 participants, do not forget that qualitative studies are local and the characteristics of this type of study mean that by definition they cannot generalize. What makes you think this? what are they based on ?. I think it is more of a limitation than a strength.
All the best

Author Response

RESPOSTA

Ponto 1: Que contato houve entre os participantes do estudo e os pesquisadores, foi limitado pela situação de pandemia? Como foi feito? Seria interessante fazer uma seção de contexto e ambiente de estudo

Response 1: I clarify that the project started in 2018, before the covid-19 pandemic, with the aim of assessing the quality of life of patients and caregivers of cancer patients seen at a state medication dispensing referral centre. The proposal was to directly interview this public, which we did initially. With the advent of the pandemic, due to health issues, we had to change the initial proposal and started sending questionnaires by e-mail and messaging applications (whatsapp). The project obtained new approval by the ethics committee due to the change in the form of data collection (amendment to the original project). The initial interviews were not used in the study because the size of the sample collected was not representative. In this sense, the questionnaires and the informed consent form were sent to the participants and filled out, when they accepted to participate in the study, and returned electronically through GOOGLE forms.
The suggestion to include section with context and study setting will be made and we understand that it will clarify this issue.

Point 2: When collecting data, they talk about questionnaires and semi-structured questions. What questions did they ask? What criteria did they use to make them? How did you request clarification if there was no face-to-face contact with the participants? As stated, there was no two-way communication as could occur in an interview.

Response 2:We used a questionnaire that collected socio-demographic data (age, education, religion, marital status, gender, income, number of people living in the same dwelling), a questionnaire on quality of life (SF-36) and a questionnaire with open questions (how do you stay active in the pandemic, what is the main concern in the pandemic, physical and emotional health, can you meet the health rules and the desire when the pandemic is over). By decision of the research group, we separated the answers to the SF-36 for a new publication of quantitative research type. As presented in the response to point 1, contact with the participants took place remotely and anonymity was obeyed. 
 Point 3: Describe in more detail guaranteed the quality of the study. How they did it?

 The research arose from the actions of initial and in-depth immersion in the environment, the stay in the field and the collection and analysis of data. This procedure supported the researcher in the selection of the
This procedure supported the researcher in the selection of the sample and the preparation of the instruments to be applied to record the data to be collected. We chose the phenomenological study because the pandemic event was new, seeking to follow the changes experienced in real situations by the participants. In this sense, the quality of the study was ensured in the selection of participants, with the participation of external auditors for monitoring and validation of the selection process. The auditor separately carried out the control and checking of data, avoiding duplicity because there were participants with multiple registrations in the institution and avoiding the inclusion of deceased/unwanted participants with active registration in the institution. Also to ensure the quality of the study, the content analysis of the speeches were categorised using MAXQDA software, which categorised the relevant information using coding and made it possible to quantify the results and calculate statistical frequencies. We used the Standards for Reporting Qualitative Research (SRQR) checklist for the presentation of results, as recommended for reporting qualitative research. 

Point 4: I find it interesting that they have the approval of the ethics committee to carry out the study (August 2018) before declaring the pandemic, please clarify this information.

Response: The research involved human beings and met the ethical-legal precepts regulated by the National Health Council of Brazil, according to Resolutions No. 466/2012 and No. 510/2016. It was submitted to the Research Ethics Committee of the Federal University of Goiás and approved on August 20, 2018 under opinion no. 2,831,905 - CAAE 93238318.7.0000.5083.  With the advent of the pandemic, with the research in progress, we had to change the data collection and subsequent amendment to the original project was submitted and approved on 25 February 2021, opinion no. 4,558,046 (CNS, 2012; CNS, 2016).

Point 5: How did you calculate the sample size?

Resposta: A seleção dos participantes incluiu usuários cadastrados no serviço de referência estadual para dispensação de medicamentos oncológicos em 2020. Foram convidados pessoalmente e por telefone nos meses de março, julho e novembro de 2020 (3 convites sucessivos), obtendo a amostra por saturação em março 2021. Os convites pessoais foram feitos para esclarecer os objetivos do estudo e garantir o sigilo dos dados, principalmente. Devido ao anonimato, não foi possível saber quem respondeu ou não, portanto, visando o maior número de participantes, foram feitos três convites com intervalos de tempo significativos. Até março de 2021, ou seja, um ano de coleta de dados, obtivemos 42 participantes e esse público contemplou nossa amostra.

Ponto 6: Diante dos resultados, o tema da pandemia dificilmente aparece, apenas no último tópico, nos três primeiros nenhuma menção é feita. Além disso, o último tópico é abordado brevemente e, portanto, não tem tanto destaque a ponto de destacá-lo dentro do título. Encontre um título que melhor se adapte ao conteúdo do artigo. Da mesma forma, os resultados obtidos desviam-se do objetivo geral do estudo, seria interessante se houvesse uma correspondência proporcional entre os resultados e os objetivos traçados.

Response:I believe the whole article addresses the issue of pandemic COVID-19, however I will discuss with the research group to alter some points in the results to positively reinforce this issue.The overall objective of the study was to analyse the perception of self-care, concerns and attitudes of caregivers of people with cancer in the face of the global pandemic of COVID-19. In this sense, we obtained that they do not practice regular physical activities (self-care), worry about their own and family physical and financial survival (concerns), comply with health guidelines regarding social distancing, use of masks and routine hand hygiene, and wish to strengthen ties of affection when the pandemic ends (attitudes). We believe that there was correspondence between the general objective and results.

Point 7:Na discussão, afirmam que os resultados poderiam ser generalizados, mesmo partindo de um tamanho de amostra de 42 participantes, não se esquecendo que os estudos qualitativos são locais e as características desse tipo de estudo fazem com que por definição não possam generalizar. O que faz você pensar isso? em que são baseados? Acho que é mais uma limitação do que uma força.

Response:This topic obtained a lot of disagreement among the members of the research group. Considering many qualitative studies in this area of cancer caregivers with much smaller sample sizes, we thought that the number of 42 could be considered relatively high by comparison. We will remove this statement from the study as suggested. We thank you for your valuable contributions.

Reviewer 2 Report

Since this is being read by people around the world it would be nice to see the actual form used. 

You mention that the predominant religion was Catholic in one place and give numbers in the table. Only three categories are listed. I am left wondering if you asked only about those three or if those were the only responses.

Including the the form would clarify these types of questions.

The discussion section does not discuss effects of what is listed as the identified themes. I would suggest rewriting the discussion section based on the identified themes and the effects on the caregivers.

Author Response

Response to Reviewer 2 Comments:

Point 1: You mention that the predominant religion was Catholic in one place and give numbers in the table. Only three categories are listed. I am left wondering if you asked only about those three or if those were the only responses. 

Including the the form would clarify these types of questions.

Response 1: I thank you for your valuable comments. I clarify that the types of religions that were presented were the three only, so they were in the survey. All responses were included in the paper. Regarding the questionnaires, we intend to include them as supplementary material to be made available to readers who request them until the entire ongoing research project is completed. After completion, the materials will be deposited in the FIGshare repository preferably.

Point 2: The discussion section does not discuss effects of what is listed as the identified themes. I would suggest rewriting the discussion section based on the identified themes and the effects on the caregivers.

Response 2: The suggestion will be discussed with the other authors and the project coordinator. Thank you.

Round 2

Reviewer 1 Report

I thank the authors for the opportunity to review the work again, I believe that certain methodological aspects have now been improved and for my part the article could be accepted

Reviewer 2 Report

The updates improved the overall understanding of the implications of the findings. Thank you for the revision.